# Comparing Different Coupling and Modeling Strategies in Hydromechanical Models for Slope Stability Assessment

Shirin Moradi [1,*], Johan Alexander Huisman [1], Harry Vereecken [1] and Holger Class [2]

1   Agrosphere Institute (IBG 3), Forschungszentrum Jülich GmbH, 52428 Jülich, Germany; s.huisman@fz-juelich.de (J.A.H.); h.vereecken@fz-juelich.de (H.V.)
2   Institute for Modeling Hydraulic and Environmental Systems (IWS), University of Stuttgart, 70569 Stuttgart, Germany; holger.class@iws.uni-stuttgart.de
*   Correspondence: s.moradi@fz-juelich.de

**Abstract:** The dynamic interaction between subsurface flow and soil mechanics is often simplified in the stability assessment of variably saturated landslide-prone hillslopes. The aim of this study is to analyze the impact of conventional simplifications in coupling and modeling strategies on stability assessment of such hillslopes in response to precipitation using the local factor of safety (LFS) concept. More specifically, it investigates (1) the impact of neglecting poroelasticity, (2) transitioning from full coupling between hydrological and mechanical models to sequential coupling, and (3) reducing the two-phase flow system to a one-phase flow system (Richards' equation). Two rainfall scenarios, with the same total amount of rainfall but two different relatively high (4 mm h$^{-1}$) and low (1 mm h$^{-1}$) intensities are considered. The simulation results of the simplified approaches are compared to a comprehensive, fully coupled poroelastic hydromechanical model with a two-phase flow system. It was found that the most significant difference from the comprehensive model occurs in areas experiencing the most transient changes due to rainfall infiltration in all three simplified models. Among these simplifications, the transformation of the two-phase flow system to a one-phase flow system showed the most pronounced impact on the simulated local factor of safety (LFS), with a maximum increase of +21.5% observed at the end of the high-intensity rainfall event. Conversely, using a rigid soil without poroelasticity or employing a sequential coupling approach with no iteration between hydromechanical parameters has a relatively minor effect on the simulated LFS, resulting in maximum increases of +2.0% and +1.9%, respectively. In summary, all three simplified models yield LFS results that are reasonably consistent with the comprehensive poroelastic fully coupled model with two-phase flow, but simulations are more computationally efficient when utilizing a rigid porous media and one-phase flow based on Richards' equation.

**Keywords:** rainfall induced landslide; slope stability; poroelasticity; local factor of safety; coupled hydromechanical modeling

## 1. Introduction

It is well known that many landslides are triggered by rainfall [1–5]. When rainwater infiltrates the soil, it alters both the overall weight and pore pressure, subsequently impacting how stress is distributed within hillslopes. This underscores the notion that rainfall-induced landslides are a prominent example of a hydromechanical process. Such interconnected hydromechanical phenomena have been understood for a considerable time, as seen in earlier studies, e.g., [6]. In recent decades, a range of models has been developed to evaluate the stability of hillslopes by taking such subsurface hydromechanical processes into account [7–11].

Existing hydromechanical models often rely on a series of simplifications. Below, we briefly review the most widely used and commonly accepted simplifications. Firstly, in most slope stability analysis models, the soil is treated as a rigid medium, neglecting

the dynamic interplay between hydraulic and mechanical factors [12–14]. For instance, while there has been empirical work on characterizing the impact of variable effective stress on porosity, e.g., [15,16], only a handful of studies have explored this potential hydromechanical interplay in stability assessments, e.g., [17,18]. In many cases, the focus has been primarily on the effects of mechanical compaction and consolidation due to increased self-weight with depth or external loading, with typical parameterizations reflecting diminishing porosity and hydraulic conductivity with depth, e.g., [9,19]. However, such approaches commonly overlook the dynamic variations in these properties.

Secondly, hydromechanical models used in slope stability assessments often employ a sequential coupling method, which primarily considers a one-way influence of transient hydrological conditions (e.g., pore water pressure) on the mechanical properties of variably saturated hillslopes. Specifically, this approach involves solving the mass and momentum balance equations for subsurface flow first and then utilizing the resulting pressure head and bulk density distribution as inputs for solving the momentum balance equations for the mechanical aspect, accounting for the corresponding suction and effective stresses within the variably saturated porous medium, e.g., [7,20,21].

Thirdly, it is common practice to assume constant pore air pressure and to simplify the actual two-phase (air and water) flow dynamics [22] into a single-phase (water) flow model where Richards' equation [23] is employed to simulate subsurface flow, e.g., [24,25].

All of the aforementioned simplifications are aimed at reducing computational costs and addressing issues related to numerical robustness, thereby enhancing the efficiency of slope stability assessment using hydromechanical models, e.g., [20,26]. However, it is important to note that these simplifications can potentially compromise model accuracy. For instance, it is demonstrated that a sequential coupling strategy lacking feedback from mechanical processes to hydraulic properties can introduce a significant error when modeling aquifer subsidence, especially when compared to a more sophisticated iterative coupling approach [26] that considers this interaction to some extent. Additionally, Cho [21] found that stability analysis results for partially saturated slopes could differ by over 10% between one-phase and two-phase flow systems.

To address inaccuracies stemming from neglecting the interplay between variable mechanical parameters and hydraulic properties, a viable solution is the adoption of a fully coupled hydromechanical model [27]. In a fully coupled approach, the mass and momentum balance equations for subsurface flow and soil mechanics are simultaneously solved within each simulation time step. Alternatively, one can opt for a sequentially coupled hydromechanical model [28], where the unknowns related to flow and soil mechanics are solved sequentially. This sequential approach can involve varying numbers of iterations between sub-problems at each time step, e.g., [29], ensuring that the influence of altered mechanical parameters on soil hydraulic properties is considered in subsequent iterations of the same time step. In principle, simulations using a sequentially coupled model with iterations, continued until the mass and momentum balance solutions converge, should yield results identical to those from an equivalent fully coupled model, e.g., [30,31]. Notably, recent developments in fully coupled hydromechanical modeling, as seen in Darcis [32], have expanded its applicability. Beck et al. [30] extended Darcis' [32] model and introduced sequential coupling with iterations. This extended model accounts for two-phase flow and incorporates variations in hydraulic parameters due to the elastic deformation of porous media caused by transient pore pressure changes. This typically occurs when the advancing saturation front encounters initially drier soil. The results of Beck et al. [30] emphasize that significant disparities can arise between non-iterative sequential and fully coupled models, particularly in highly transient conditions featuring substantial gradients in pore water pressure and stress. These disparities are more pronounced when computational resources limit the number of feasible iterations in the sequentially coupled model. In recent years, the adoption of fully coupled hydromechanical models has grown across various applications, including land deformation, water table determination [33], hydraulic fracturing [34,35], clay activities [36], and reservoir characterization [37]. However, such comprehensive modeling

strategies have not been widely employed for slope stability assessment [38–42]. Furthermore, the errors associated with widely used model simplifications in the assessment of variably saturated hillslopes have yet to be thoroughly examined.

Within this context, the aim of this study is to compare various coupling and modeling strategies for assessing the stability of variably saturated hillslopes using hydromechanical models. Additionally, we seek to assess the associated errors resulting from different coupling approaches and commonly employed model simplifications. To achieve this, we made modifications to both the fully coupled hydromechanical model developed by Darcis [32] and the non-iterated sequential model proposed by Beck et al. [30]. These modifications enabled us to simulate variations in water content and stress distribution within variably saturated hillslopes. Furthermore, we simplified the two-phase fully coupled model in two ways: first, into a two-phase fully coupled model featuring a rigid soil with constant porosity and no poroelasticity, and second, into a fully coupled model incorporating a one-phase flow system (Richards' model) for water flow (refer to Supplement S1, Equations (S1)–(S19) for details). We then evaluated the impact of these different coupling and modeling strategies on simulated slope stability for different rainfall-induced infiltration conditions up to the point of failure. Our assessment was carried out using the Local Factor of Safety method [7].

## 2. Materials and Methods

### 2.1. Coupled Hydromechanical Model

The key elements of the hydromechanical modeling approach have been described [30,43], and readers are referred to the Supplementary Material Index S1 for more details.

### 2.2. Evaluation of Stability Status

Once the mass and momentum balance equations have been solved (either simultaneously or sequentially), the stability of the variably saturated hillslope can be evaluated. In this study, the Local Factor of Safety (LFS) approach proposed by Lu et al. [7] has been used, which can best be implemented for early warning of failure initiation. The LFS [−] is based on the Mohr–Coulomb criterion (Figure S2) and is the ratio of Coulomb stress at the potential failure state, $\tau^*$ [$ML^{-1}T^{-2}$], and the current state of Coulomb stress, $\tau$ [$ML^{-1}T^{-2}$], at each point within a hillslope. LFS = 1, therefore, defines the stability threshold, where failure potentially occurs for values lower than 1.0. The Coulomb stress at the potential failure state, $\tau^*$, can be defined by

$$\tau^* = c' + \sigma' \tan \phi' \tag{1}$$

in which $c'$ [$ML^{-1}T^{-2}$] is the effective cohesion, $\sigma'$[$ML^{-1}T^{-2}$] is the effective stress, and $\phi'$ [°] is the effective internal friction angle of the soil. The LFS of each element within the hillslope is then calculated as

$$\text{LFS} = \frac{\tau^*}{\tau} = \frac{\cos \phi' \, (c' + \sigma_I' \tan \phi')}{\sigma_{II}'} \tag{2}$$

where $\sigma_I'$[$ML^{-1}T^{-2}$] and $\sigma_{II}'$[$ML^{-1}T^{-2}$] are obtained based on the maximum and minimum principal stresses, $\sigma_{1,3}$ [$ML^{-1}T^{-2}$], and the suction stress, $\sigma^s$[$ML^{-1}T^{-2}$], as

$$\sigma_I' = \frac{\sigma_1 + \sigma_3}{2} - \sigma^s \tag{3}$$

$$\sigma_{II}' = \frac{\sigma_1 - \sigma_3}{2} \tag{4}$$

One can reasonably infer that alterations in the size of the Mohr circle, delineated by the principal total stresses and influenced by the bulk weight and element positioning within a hillslope, remain relatively modest in a variably saturated hillslope undergoing infiltration.

However, cumulative pore water pressure in an unsaturated hillslope reduces the effective stress's absolute value, causing the Mohr circle to shift leftward (refer to Figure S2), nearing the failure envelope. Consequently, pore pressure primarily determines the Mohr circle's position relative to the Mohr–Coulomb failure envelope [7,44]. Initial Local Factor of Safety (LFS) results align with conventional stability assessment methods [7,45]. However, this approach offers additional insights into when and where failure might initiate, eliminating the need to predefine a failure surface. The LFS method has previously been used with a sequentially coupled hydromechanical model (without iterations) and a one-phase (water) flow system [7,12,46].

### 2.3. Implementation of Different Coupling and Modeling Concepts

Four different implementations of coupled hydromechanical models with the LFS concept are compared (Table 1). The most comprehensive implementation is based on Darcis' [32] fully coupled hydromechanical model with two-phase flow. It considers the influence of pore pressure and elastic volumetric strain on porosity and hydraulic conductivity. The second implementation simplifies Darcis [32] model by excluding poroelastic effects. The third model is based on the sequentially coupled model of Beck et al. [30] with no iterations between the hydrological and mechanical parts. Feedback from the mechanical model to the hydrological model occurs in the next time step instead of the same time step. The fourth and final implementation simplifies the fully coupled two-phase flow model to a fully coupled model with one-phase flow (Richards' equation).

**Table 1.** Four different implementations of coupled hydromechanical models in this study.

| Model | Abbreviation |
| --- | --- |
| Fully coupled two-phase flow model with variable porosity | 2P-FC-var.Por. |
| Fully coupled two-phase flow model with constant porosity | 2P-FC-const.Por. |
| sequentially coupled two-phase flow model | 2P-SC |
| One-phase flow model (Richards' equation) | 1P-FC |

All four model implementations were realized in DuMux [27,47,48], which is a free, open-access "multi- [physics, . . .]" simulator of fluid flow in porous media (https://dumux.org/ (accessed on 1 November 2023)). DuMux is based on the Distributed and Unified Numerics Environment (DUNE) [49–51] and solves the partial differential equations (PDE) for fluid flow and soil mechanics using a finite volume method. The model domain is discretized using the Box method [27].

To compare the four model implementations, the simulations were conducted on a 2D sloped domain with a 30° incline, featuring idealized isotropic and homogeneous silty soil (Figure 1). The stability of this failure-prone slope has been previously assessed using the LFS concept [7,12,46]. Hydromechanical soil properties are detailed in Table S1, and boundary conditions are depicted in Figure 1. For the hydrological model, a no-flow condition at the bottom and left boundaries of the domain were applied. Initially, the slope was assumed to be in hydrostatic equilibrium with a 5 m deep groundwater table extending horizontally from the toe of the slope. The unknown primary variables of the flow equation in the so-called fully coupled approach are pore water pressure ($p_w$) and air saturation ($S_a$). Hence, the right boundary below the groundwater table, characterized by full saturation ($S_a = 0$) and a prescribed hydrostatic water pressure ($p_w$), is treated as a Dirichlet boundary. Above the water table, we set a no-flow boundary. The top surface acts as a Neumann boundary with an infiltration rate matching the rainfall rate. Here, two constant rainfall intensities were employed. The first set of simulations used a low-intensity rainfall (LIR) of 1 mm h$^{-1}$ (20% of $K_s$) for 20 h. The second set applied a high-intensity rainfall (HIR) of 4 mm h$^{-1}$ (80% of $K_s$) for 5 h to ensure an equivalent total infiltration amount for both events. In the mechanical model, the top surface is a free boundary, while the left and right boundaries have no displacement in the direction normal to the boundary (roller boundary). The bottom is fixed.

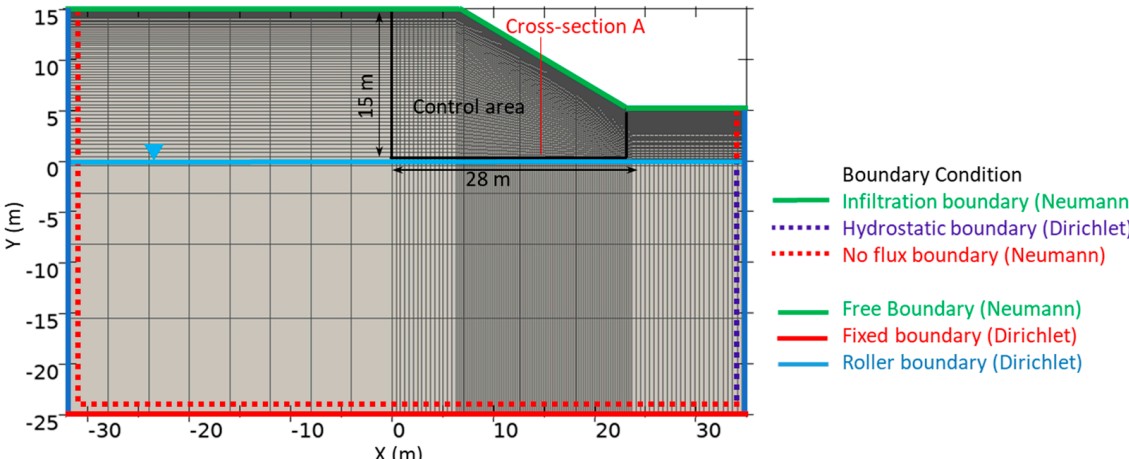

**Figure 1.** The geometry, boundary conditions, and discretization of the 2D, homogeneous silty slope used to compare the four hydromechanical model implementations. Please note that the same color and line type were used for the hydrological infiltration and mechanical free boundary conditions at the surface because they are both of the Neumann-type.

A cube mesh was used to discretize the modeling domain (Figure 1) with an increasing grid size and depth to reduce the computational costs while being able to catch the hydromechanical dynamics that are expected to be most pronounced near the slope surface [12]. The following analysis is focused on a relatively small part of the modeling domain indicated as the control area in Figure 1 and the associated cross section A. This control area was used to reduce the effect of the boundary conditions on the seepage condition of the slope, e.g., [52]. As reported by Kristo et al. [53], setting the side boundaries at a distance of 3 times the height of the slope from the crest and the toe minimizes the impact of the boundary conditions on the simulation results.

## 3. Results

### 3.1. Fully Coupled Two-Phase Flow Model with Variable and Constant Porosity

Figure 2 depicts the evolving LFS distribution within the 2D slope, simulated using the comprehensive fully coupled two-phase flow model under both low- and high-intensity rainfall events. This model accounts for poroelastic effects linked to variable pore pressure and material self weight. Figure 2 shows that the LFS gradually diminishes near the slope surface as infiltration progresses, with the potentially unstable area, characterized by an LFS near 1.0, initially emerging near the slope toe and expanding over time. The LFS method considers only the shear stress of each element independently, without regard for neighbouring elements, and it does not account for post-failure stress redistribution. Consequently, an LFS < 1 does not unequivocally denote a state of failure [43], and the slope stability status after the initial potentially unstable location should be interpreted cautiously. Therefore, all simulation results are presented until the LFS reaches the potential failure threshold of LFS = 1.0 at some point on the hillslope.

To investigate the significance of incorporating poroelasticity in slope stability assessment, the dynamics of vertical effective stress for the two rainfall intensities, focusing on cross section A of the hillslope are presented in Figure 3. This figure displays the simulated effective stress as a function of depth relative to the effective stress derived from bulk density and the hydrostatic pore pressure distribution, assuming a constant porosity of 0.46. The resulting changes in simulated porosity are illustrated in Figure 4. The simulations reveal that the effective stress was lower than the specified value above a depth of approximately 5 m, but it exceeded this value below this depth. A positive change in effective stress signifies that the compressive pressure due to self-weight exceeded the specified value. Rainfall infiltration also led to a reduction in effective stress and an increase in porosity near the surface. However, the overall changes in simulated porosity were

relatively minor, with a maximum shift of +0.59% observed near the slope surface after the HIR and +0.58% during the LIR. The average difference at cross section A was only 0.08% for the upper 7 m and 0.37% for the upper 1 m for both rainfall intensities. Subsequently, the focus narrows to near-surface pore water pressure and LFS dynamics due to rainfall infiltration, with simulation results presented solely for the upper 1 m. All differences are presented as relative values compared to the outcomes of the comprehensive fully coupled model (i.e., $(X_x - X_{FC})/X_{FC} \times 100$), with a positive change indicating an increase relative to the equivalent value in the comprehensive fully coupled model (FC) and a negative change representing a decrease in the respective value.

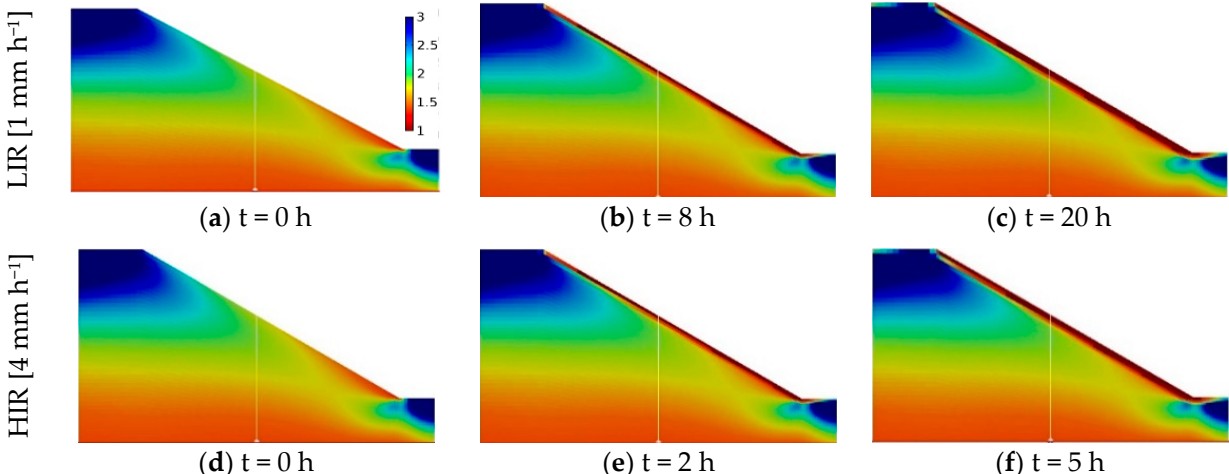

**Figure 2.** The spatial and temporal variability of LFS for the 2D silty slope simulated with the FC two-phase flow model for the LIR at (**a**) t = 0 h, (**b**) t = 8 h, and (**c**) t = 20 h, and the HIR at (**d**) t = 0 h, (**e**) t = 2 h, and (**f**) t= 5 h.

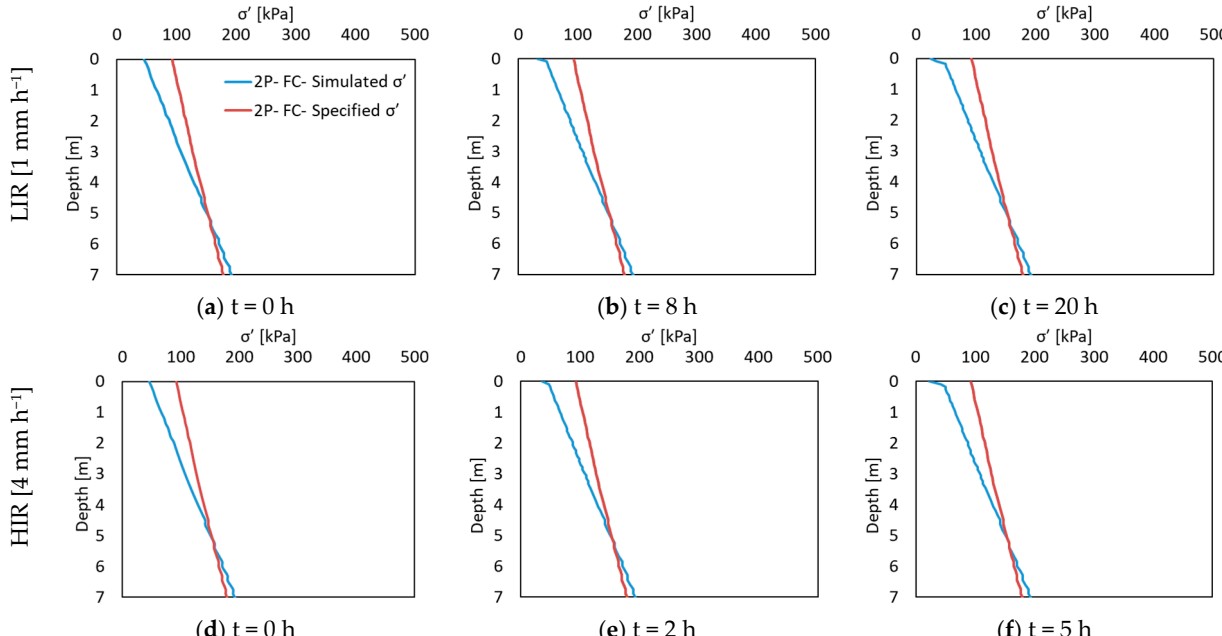

**Figure 3.** The simulated change in effective stress at cross section A of the 2D silty slope using the two-phase FC model for the LIR at (**a**) t = 0 h, (**b**) t = 8 h, and (**c**) t = 20 h and the HIR at (**d**) t = 0 h, (**e**) t = 2 h, and (**f**) t = 5 h.

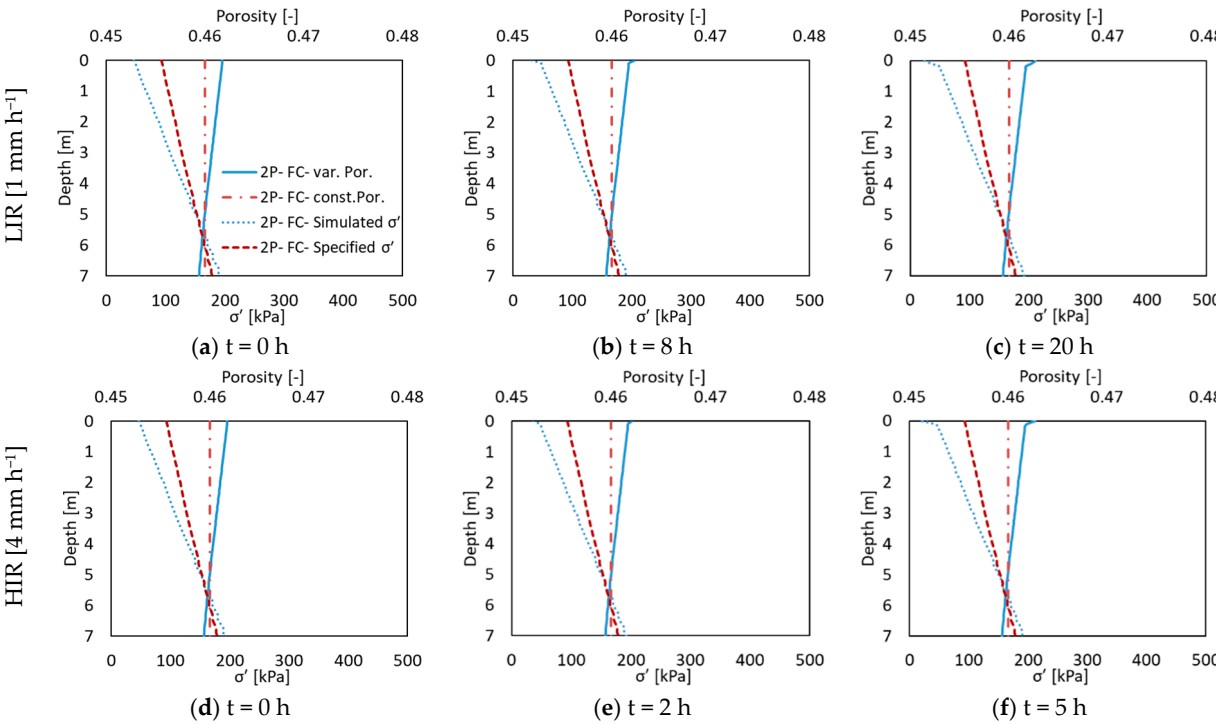

**Figure 4.** The dynamics of simulated vertical effective stress and porosity at cross section A of the 2D silty slope using the two-phase FC model for the LIR at (**a**) t = 0 h, (**b**) t = 8 h, and (**c**) t = 20 h and the HIR at (**d**) t = 0 h, (**e**) t = 2 h, and (**f**) t = 5 h.

To assess the impact of poroelasticity on pore water pressure ($p_w$) and stability, we compared results from the comprehensive fully coupled two-phase flow model with those from the same model employing a constant porosity (Figure 5). The most significant disparities in $p_w$ and LFS between the two model implementations were observed at the soil surface and at the end of the high-intensity rainfall event. Specifically, the maximum discrepancies reached approximately $-10.1\%$ for $p_w$ (decrease) and $+2.0\%$ for LFS (increase) compared to the fully coupled model with poroelasticity. In the case of the low-intensity rainfall, the variations amounted to a maximum of $-2.2\%$ for $p_w$ and $+1.1\%$ for LFS at the end of the event. When considering the entire cross section for the high-intensity rainfall event, the average differences were only $-0.8\%$ for $p_w$ (decrease) and $+0.2\%$ for LFS (increase). Similarly, at the end of the low-intensity rainfall, these values were $-0.3\%$ for $p_w$ and $+0.1\%$ for LFS.

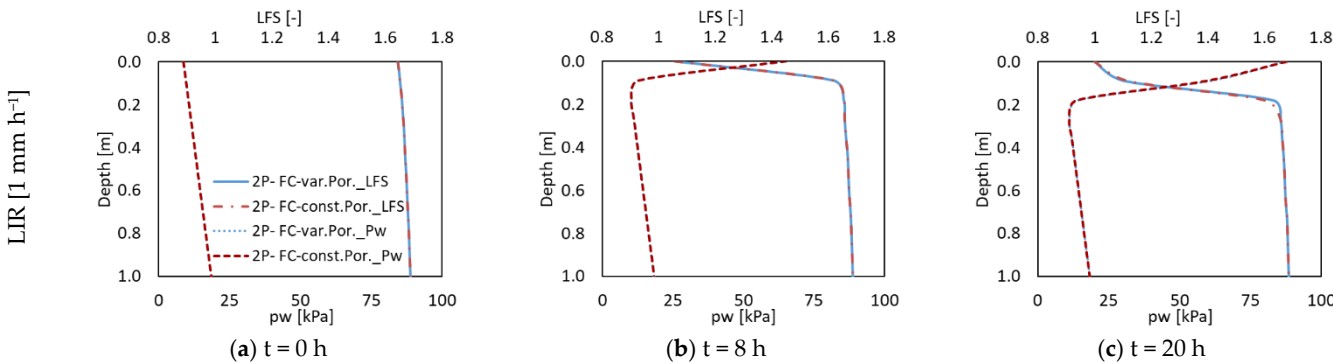

**Figure 5.** *Cont.*

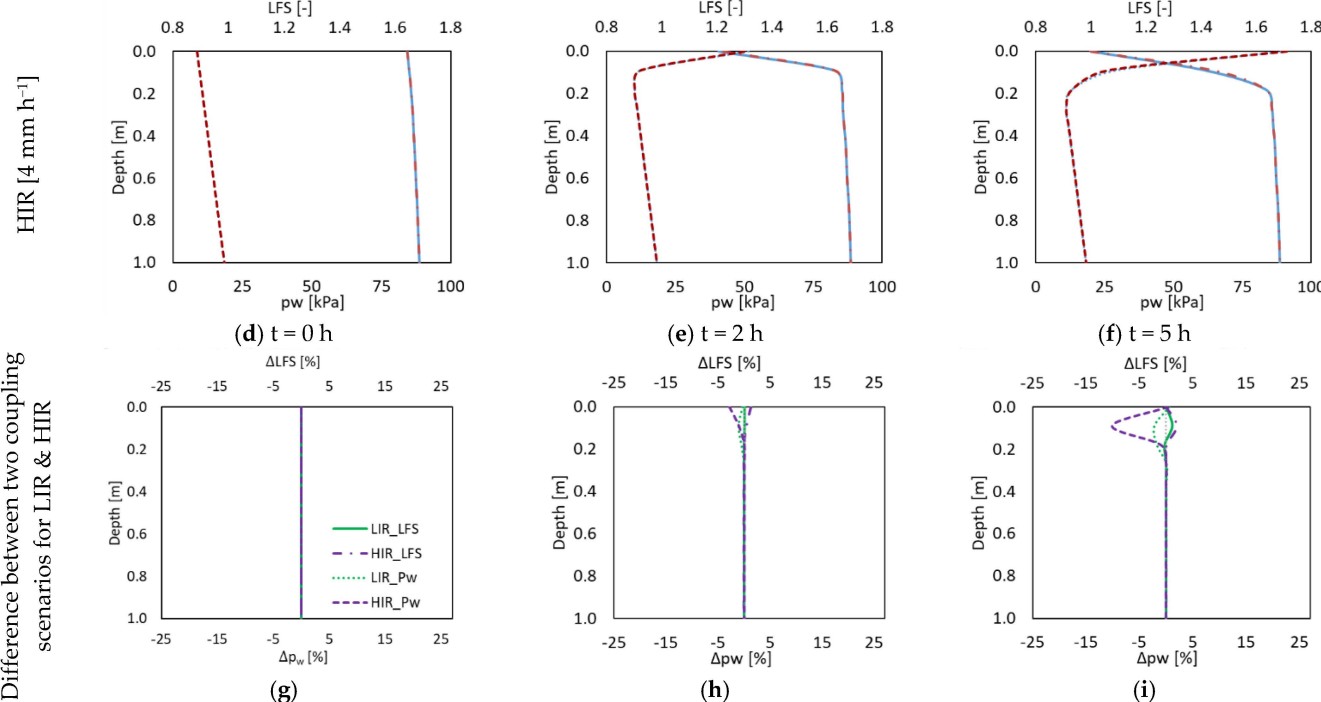

**Figure 5.** The simulated $p_w$ and LFS distribution at cross section A of the 2D silty slope using the two-phase fully coupled (2P-FC) model with variable and constant porosity for the LIR at (**a**) t = 0 h, (**b**) t = 8 h, and (**c**) t = 20 h and the HIR at (**d**) t = 0 h, (**e**) t = 2 h, and (**f**) t = 5 h. The difference between the two model implementations for low- and high-intensity rainfall are shown in panels (**g**–**i**).

### 3.2. Fully Coupled vs. Sequentially Coupled Models

To compare simulation outcomes between the fully coupled (FC) and sequentially coupled (without iterations) two-phase flow models, we examined simulated pore water pressure ($p_w$) in Figure 6 for the upper 1 m of cross section A during both low- and high-intensity rainfall events. The results reveal a maximum variation of −16% in $p_w$ with the sequentially coupled (SC) model during the high-intensity rainfall event, occurring in the middle of the event. In Figure 6, we also illustrate the resulting differences in simulated LFS. As the initial conditions for all models were identical, encompassing the same initial pressure distribution and LFS, these aspects are not depicted in subsequent figures. The sequentially coupled model exhibited a +7.5% deviation compared to the fully coupled model, with the most significant difference also appearing in the middle of the high-intensity rainfall event. For the low-intensity rainfall event, the sequentially coupled model showed a maximum variation of −6.3% in $p_w$ and a +4.3% difference in LFS. Averaging the top 1 m of cross section A during the high-intensity rainfall event, we observed average differences of −1.5% for simulated $p_w$ and +0.3% for LFS. Corresponding averages during the low-intensity rainfall event were −0.4% for $p_w$ and +0.2% for LFS. These disparities predominantly occurred near the surface, where dynamic changes in pore water pressure were most pronounced. For depths exceeding 1 m, the effect of increased soil weight due to infiltration remained below 0.01% for both $p_w$ and LFS.

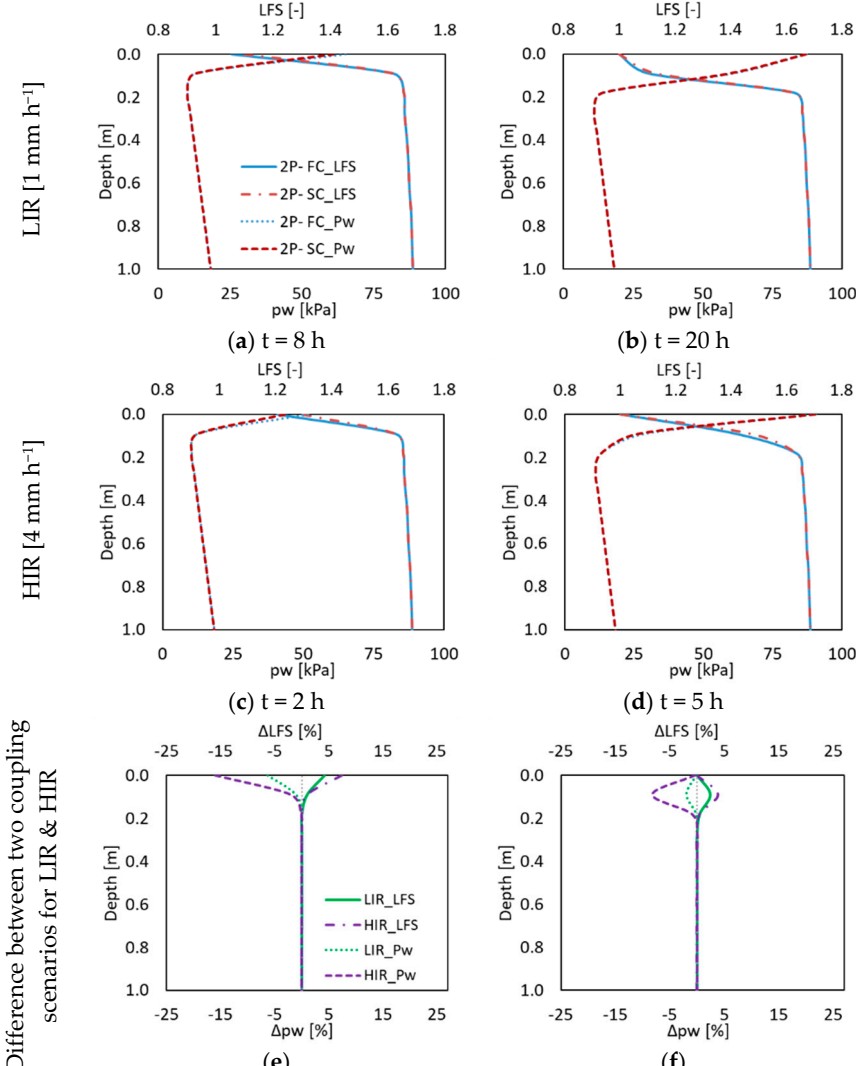

**Figure 6.** The simulated $p_w$ and LFS distribution at cross section A for the 2D silty slope using fully and sequentially coupled two-phase flow models (2P-FC and 2P-SC, respectively) with the LIR at (**a**) t = 8 h and (**b**) t = 20 h and the HIR at (**c**) t = 2 h and (**d**) t = 5 h. The differences between the two model implementations for low- and high-intensity rainfall are shown in panels (**e**,**f**).

### 3.3. Fully Coupled Two-Phase vs. One-Phase Flow Model (Richards' Equation)

Finally, we compare the simulated results between the fully coupled two-phase flow model (2P-FC) and the fully coupled one-phase flow model (Richards' equation) (1P-FC). Figure 7 illustrates the simulated pore water pressure ($p_w$) and LFS for both model implementations in the upper 1 m of cross section A during the low- and high-intensity rainfall events, along with their relative differences. Once more, the most significant deviation between the two models arises at the end of the high-intensity rainfall event. Here, we observe a +97.2% shift in $p_w$ and a −21.5% shift in LFS compared to the fully coupled model. This maximum difference diminishes to roughly +53.7% in $p_w$ and −11.9% in LFS for the low-intensity rainfall event. Additionally, we notice that the disparities between the two model implementations primarily occur in a smaller region near the slope surface during the high-intensity rainfall event. Consequently, the average differences for the top 1 m of cross section A are relatively consistent (−9.5% for $p_w$ and 1.9% for LFS at the end of the high-intensity rainfall and −8.2% for $p_w$ and 1.9% for LFS at the end of the low-intensity rainfall event, relative to the fully coupled model).

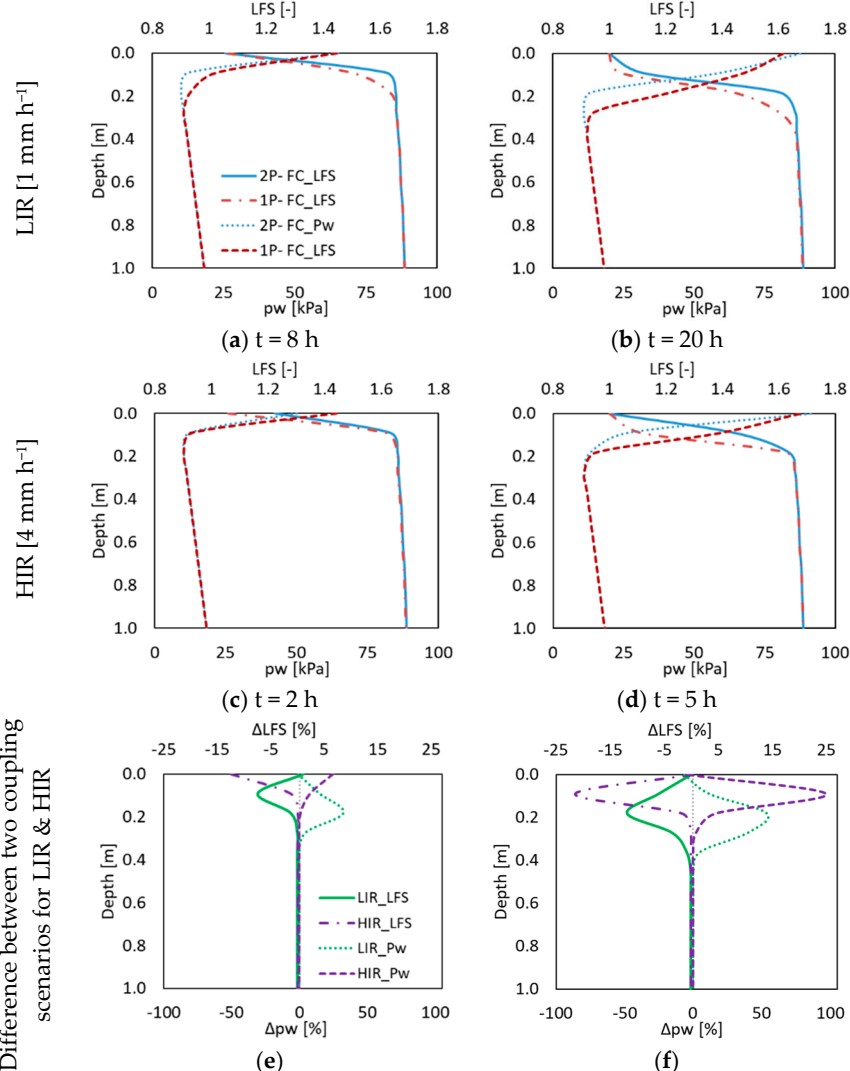

**Figure 7.** The simulated $p_w$ and LFS distribution at cross section A for the 2D silty slope using the fully coupled two-phase and Richards' models (2P-FC and 1P-FC, respectively) with the LIR at (**a**) t = 8 h, and (**b**) t = 20 h and the HIR at (**c**) t = 2 h, and (**d**) t = 5 h. The differences between the two model implementations for low- and high-intensity rainfall are shown in panels (**e**,**f**).

## 4. Discussion

Table 2 summarizes the results of the aforementioned modeling simplifications and coupling strategies on both the pore water pressure and stability assessment of variably saturated hillslopes, which is used as a starting point for the following discussion.

**Table 2.** Comparative analysis of simulated pore water pressure and Local Factor of Safety of the simplified models relative to the comprehensive fully coupled model (in percentage) under the two investigated rainfall intensities.

| 2P-FC-var.Por. vs. . . . | Parameter | HIR (4 mm h$^{-1}$) (%) | LIR (1 mm h$^{-1}$) (%) |
|---|---|---|---|
| 2P-FC- const-Por. | $p_w$ | −10.1 | −2.2 |
| | LFS | +2.0 | +1.1 |
| 2P-SC | $p_w$ | −16.0 | −6.3 |
| | LFS | +7.5 | +4.3 |
| 1P-FC | $p_w$ | +97.2 | +53.7 |
| | LFS | −21.5 | −11.9 |

### 4.1. Effect of Poroelasticity

In a poroelastic medium, porosity changes with depth and pore water pressure. Where pore pressure exceeds the bulk weight, porosity increases; where self weight surpasses pore pressure, porosity decreases. Thus, areas with positive changes in vertical effective stress tend to have decreasing porosity, as seen in Figure 3. Conversely, areas with negative changes in effective stress are expected to exhibit increased porosity.

The results in Figure 4 suggest that the increased effective hydraulic conductivity ($K_{eff}$) due to higher porosity in the poroelastic model outweighs the displacement ($u$) effect, resulting in elevated pore pressure near the surface. In contrast, simulations with constant hydraulic conductivity in a poroelastic fully coupled model showed lower pore pressure (results not shown) and consequently higher LFS at those locations.

The maximum model differences coincide with those of the transient zones, which is consistent with the findings of Beck et al. [30]. These zones are located near the surface and experience maximum pore pressure variations during the wetting front infiltration. For different infiltration rates, the pore water pressure gradient differs across the infiltration front. In particular, the pore water pressure was higher for the high-intensity rainfall, which resulted in a stronger increase in porosity. For this reason, the effect of not considering poroelastic effects is more pronounced in the case of high-intensity rainfall. The higher pore water pressure resulted in lower LFS in the simulated results of the model that considers poroelasticity. Overall, the influence of poroelasticity and its effects on pore water pressure and stability in variably saturated hillslopes appear relatively minor during infiltration.

### 4.2. Effect of Coupling Strategy

For both rainfall intensities, the slightly higher $p_w$ values in the fully coupled model are due to its consideration of the complete interaction between effective pore pressure and volumetric strain within each time step. In the sequentially coupled model with no iteration, the impact of variable pore pressure on stress and strain distribution occurs in the same time step, but the feedback of volumetric strain to pore pressure is accounted for in the subsequent time step. As previously discussed, differences are more prominent during high-intensity rainfall events due to steeper pore water pressure gradients. These results again align with the findings of Beck et al. [30], emphasizing that the maximum difference between fully coupled and sequentially coupled models occurs in regions with transient processes involving rapid pore water pressure changes within a single time step. Despite the anticipated greater reliability of the fully coupled model due to its comprehensive interaction between hydrological and mechanical components, the results presented here indicate relatively minor variations in pore water pressure and consequent instability assessment between these two coupling strategies (Table 2).

Notably, the differences in simulated $p_w$ and LFS between fully and sequentially coupled models are more significant than those between the fully coupled model with and without poroelastic changes. This smaller difference in the latter simulations is attributed to the counteractive effects of displacement ($u$) and effective hydraulic conductivity ($K_{eff}$) on pore pressure in the fully coupled poroelastic model. Specifically, increased effective porosity due to infiltration reduces pore pressure and elevates $K_{eff}$. The latter increases pore pressure, partially offsetting the overall effect of larger pore size, resulting in a reduced pore pressure decrease or even an increase.

### 4.3. Effect of the Multiphase Flow Model

The results from comparing the fully coupled two-phase and one-phase flow models indicate a more significant disparity between these two implementations than the impact of simplifying poroelasticity or employing different coupling strategies. These differences in Figure 6 are primarily attributed to the influence of constant pore air pressure ($p_a$) in the one-phase flow model based on Richards' equation. In a two-phase flow system involving water and air, the air must move or exit the domain during rainfall infiltration. Consequently, the downward progression of water is impeded by elevated pore air pressure,

resulting in reduced flow and a slower increase in $p_w$. This also explains the enhanced water accumulation near the surface over time, leading to higher $p_w$ near the surface in the two-phase flow model. To substantiate this, we analyzed the development of pore air pressure ($p_a$) during both low- and high-intensity rainfall events. Figure 8 illustrates the rising trend of air pressure with infiltration. As depicted, air pressure increases with depth and infiltration. This pattern arises from soil compaction and the displacement of air from shallower layers due to increasing depth and infiltration, respectively. In the case of low-intensity rainfall, air movement is less restricted and can be released from the area, resulting in a less pronounced increase in $p_a$ with infiltration. The steeper curve for low-intensity rainfall represents the initially dominant influence of soil compaction, which is more prominently affected by infiltration during high-intensity rainfall.

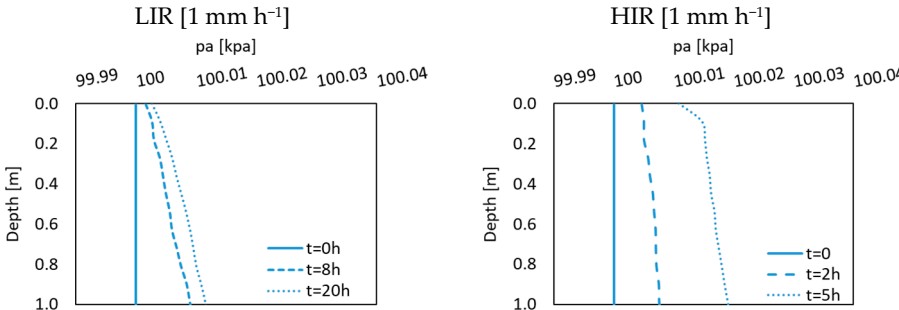

**Figure 8.** The simulated pa distribution at cross section A for the 2D silty slope using fully coupled two-phase models (2P-FC and 1P-FC, respectively) for LIR and HIR at different time steps.

The lower pore water pressure led to increased stability across various times and locations within the two-phase flow system. This enhanced stability when considering that two-phase flow is consistent with Cho [21]. During the simulations with high-intensity rainfall, the affected area is shallower and more localized compared to simulations with low-intensity rainfall despite the same total precipitation volume. Consequently, the differences between the two models are more pronounced during high-intensity rainfall, primarily within a smaller region closer to the slope surface. These results underscore the potential benefit of considering multiphase flow processes in slope stability assessment, provided sufficient computational resources are available. However, it is important to note that this study does not account for soil heterogeneity or the presence of macrospores, which are both common in natural soils. Particularly, the existence of macropores is expected to significantly enhance soil aeration, reducing the impact of air entrapment on the infiltration process.

## 5. Conclusions and Outlook

In this study, the stability status of a 2D slope was simulated using a fully coupled hydromechanical model with two-phase (water and air) flow. This model was implemented in the open-source simulator DuMux, and simulations were made for a low- and a high-intensity rainfall event. This comprehensive model was used to examine the impact of three widely used model simplifications in slope stability assessments using the local factor of safety (LFS) approach: (i) the absence of feedbacks from the mechanical to the hydrological model component, (ii) the use of a sequential instead of a fully coupled modeling approach, and iii) the use of a one-phase (i.e., water with Richards' equation) instead of a two-phase (i.e., water and air) flow model. The simulation results indicated that the most significant differences in slope stability occurred near the slope surface, where the steepest gradients in pore water pressure occurred, aligning with previous hydromechanical model applications [30].

Among the three simplifications, replacing a two-phase flow model with a one-phase flow model had the most substantial impact on simulated pore water pressure and LFS. Yet simulations were more robust and computationally efficient when utilizing rigid porous media and one-phase flow based on Richards' equation. Employing a sequentially coupled model or a fully coupled model with constant porosity caused only minor changes. However, instability primarily occurred in fully saturated areas with relatively small pore pressure variations, resulting in minimal differences between the model implementations regarding failure initiation. The disparity between the simplified models and the comprehensive fully coupled model was higher when the rainfall intensity was higher. Nevertheless, exploring different soil types with varying porosities and pore size distributions, like sand or clay with diverse fine and coarse material content, could provide further insights into infiltration patterns and pore pressure variations. In summary, all model simplifications yielded acceptable slope stability analyses, resembling outcomes from the comprehensive fully coupled two-phase flow model. Vulnerable zones prone to failure were consistent across models, offering valuable insights for slope reinforcement and early warning. However, slight variations in potential failure times suggest that more comprehensive methods may be warranted in sensitive sites with sufficient data or without computational constraints.

The results of this study can be extended by exploring more realistic slopes, accounting for heterogeneity, macropores, varying aeration conditions, transient zones, and diverse pore pressure distributions. Additionally, factors like soil types, their variability across a hillslope, groundwater levels, and slope angle play pivotal roles in determining slope stability. Boundary conditions, such as variable infiltration rates and bidirectional inflow and outflow through the boundary, should also be examined in a subsequent study. While the assumption of linear-elastic soil may suffice for early slope failure warning, future simulations can be enhanced by considering more realistic elastoplastic hillslopes, incorporating plastic deformation and post-failure stress redistribution. A three-dimensional slope with complex geometry, bedrock topography, and validation against real-world slopes may also offer further insights into the suitability of different model simplifications and coupling strategies.

**Supplementary Materials:** The following supporting information can be downloaded at https://www.mdpi.com/article/10.3390/w16020312/s1. Index S1: Theory. Figure S1: Illustration of different coupling strategies and the considered interactions between sub-problems: (a) a fully coupled model, (b) a sequentially coupled without iterations, and (c) a sequentially coupled model with iterations within each time step. Figure S2: Illustration of the local factor of safety (LFS) concept using the Mohr circle (adapted from Lu et al. [6]). Index S2: Hydraulic parameters. Table S1: Hydraulic and mechanical parameters of the simulated slope (based on Lu et al. [6]).

**Author Contributions:** Conceptualization, S.M., J.A.H. and H.C.; methodology, S.M., J.A.H. and H.C.; software, S.M., J.A.H. and H.C.; validation, S.M., J.A.H. and H.C.; formal analysis, S.M., J.A.H. and H.C.; investigation, S.M., J.A.H. and H.C.; resources, S.M., J.A.H., H.C. and H.V.; data curation, S.M., J.A.H. and H.C.; writing—original draft preparation, S.M.; writing—review and editing, S.M., J.A.H. and H.C.; visualization, S.M.; supervision, J.A.H. and H.C.; project administration, J.A.H., H.C. and H.V.; funding acquisition, J.A.H., H.C. and H.V. All authors have read and agreed to the published version of the manuscript.

**Funding:** This research was funded by the German Ministry of Education and Research (BMBF) in the framework of the R&D Program GEOTECHNOLOGIEN through the project 'Characterization, monitoring and modelling of landslide-prone hillslopes (CMM-SLIDE)' (grant number 03G08498). We also gratefully acknowledge funding by the German Ministry of Economic Affairs and Climate (BMWK) through the German Aerospace Center for the AssimEO project (grant number 50EE1914A).

**Data Availability Statement:** The data presented in this study are available on reasonable request from the authors.

**Acknowledgments:** We thank the anonymous reviewers and the associate editor for the constructive reviews that improved the quality of this manuscript.

**Conflicts of Interest:** The authors declare no conflicts of interest. The funders had no role in the design of the study; in the collection, analyses, or interpretation of data; in the writing of the manuscript; or in the decision to publish the results. Author Shirin Moradi and Johan Alexander Huisman was employed by the company Forschungszentrum Jülich GmbH. The remaining authors declare that the research was conducted in the absence of any commercial or financial relationships that could be construed as a potential conflict of interest.

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
