# Peer review of "Comparing Different Coupling and Modeling Strategies in Hydromechanical Models for Slope Stability Assessment"

_water, doi:10.3390/w16020312_

Round 1

Reviewer 1 Report

Comments and Suggestions for Authors

 The work of Moradi et al. is a numerical study. This work assesses through two cases studies the effects of simplifying the numerical implementation of hydromechanical behaviours for the evaluation of landslide hazard. This work is of good quality, nearly reaching publication standards:

·       - the state of the art is relevant.

·    -    the methodology is well described. Modifying an existing code to test the effects of different simplifications allows you to overcome potential differences linked to the numerical scheme of each code. I think this is a very good way to proceed.

·      -  The results are well described, interpreted and discussed.

·       - The conclusions remain modest and the authors recommend further relevant studies.

In order to reach publication standards, I recommend the following changes:

·       - Makes all figures bigger as well as the text of figures.

·       - Resume in a table at the beginning of part 2.3 the short names of all performed simulations. Ensure that the simulations naming is homogenous through the paper and figures (it is not the case at the moment).

·       - Always use LIR and HIR short names to describe the rainfall scenarios in the text and especially in the figures. Sometime they are also referred as 1 and 4 mm rain scenarios.

·       - in Figure 1, the representation of boundary conditions is not very easy to read. They should be represented with more different colours as well as distinct symbols.

Author Response

Dear Editor, Dear Reviewer,

We are grateful for the dedicated time and effort you invested in reviewing our manuscript. We believe that the suggested comments and revisions have enhanced the quality of the manuscript. We have diligently addressed each comment, and a detailed point-by-point response is provided below (attached). We trust that the revisions meet your expectations, and we hope for your positive consideration for the publication of this manuscript in Water.

 Best Regards,

 Shirin Moradi (on behalf of all co-authors)

Reviewer 2 Report

Comments and Suggestions for Authors

Understanding the impact of simplifications in stability assessments of landslide-prone hillslopes due to precipitation is crucial for accurately evaluating the dynamic interaction between subsurface flow and soil mechanics, aiding in more effective landslide risk mitigation strategies.

The paper investigates the simplifications commonly made in stability assessments of landslide-prone hillslopes influenced by precipitation. It focuses on three specific simplifications: neglecting poroelasticity, transitioning from full coupling to sequential coupling between hydrological and mechanical models, and reducing the two-phase flow system to a one-phase flow system. The study compares the results of these simplified approaches with a comprehensive fully coupled poroelastic hydro-mechanical model. The findings reveal that while all three simplified models provide reasonably consistent results with the comprehensive model, the most significant impact on stability assessments comes from transforming the two-phase flow system to a one-phase flow system, leading to a maximum increase of +21.5% in the local factor of safety. Conversely, neglecting poroelasticity or using a sequential coupling approach has a relatively minor effect on stability assessments. The study suggests that simplified models are computationally more efficient, especially when employing a rigid porous media and a one-phase flow based on Richards' equation.

Recommendations for improvement:

-          Explicitly justify the chosen simplifications in the stability assessment process, explaining the trade-offs between computational efficiency and accuracy, and discussing the broader applicability of the simplified models.

-          Include quantitative metrics for comparing the results of simplified models with the comprehensive model, such as statistical measures or error analysis, to strengthen the validity of the comparisons and provide a clearer basis for the conclusions.

-          Strengthen the conclusion section by explicitly summarizing the key findings and their implications for landslide stability assessments, as well as suggesting avenues for future research based on the study's limitations.

-           

-          Ensure that all references and citations are up-to-date, relevant, and properly cited throughout the manuscript. References 15, 16, 19, 24, 28, 29, 57 are out of date.

By addressing these recommendations, the authors can enhance the manuscript's overall quality, strengthen the validity of their findings, and ensure broader relevance and applicability to the scientific community.

Author Response

(The authors gave the same response as above.)
